# Rapid Generation and Molecular Docking Analysis of Single-Chain Fragment Variable (scFv) Antibody Selected by Ribosome Display Targeting Cholecystokinin B Receptor (CCK-BR) for Reduction of Chronic Neuropathic Pain

**DOI:** 10.3390/ijms241311035

**Published:** 2023-07-03

**Authors:** Adinarayana Kunamneni, Marena A. Montera, Ravi Durvasula, Sascha R. A. Alles, Sachin Goyal, Karin N. Westlund

**Affiliations:** 1Department of Internal Medicine, Mayo Clinic, Jacksonville, FL 32224-1865, USA; 2Department of Medicine, Loyola University Medical Center, Maywood, IL 60153-3328, USA; 3Department of Anesthesiology & Critical Care Medicine, University of New Mexico Health Sciences Center, Albuquerque, NM 87131-0001, USA; 4Biomedical Laboratory Research & Development (121F), New Mexico VA Health Care System, Albuquerque, NM 87108-5153, USA

**Keywords:** scFv, antibody library, ribosome display, molecular docking, chronic pain, nerve injury, neuropathy, hypersensitivity, anxiety, depression, pain therapy, cholecystokinin B, CCK, non-addictive pain therapy

## Abstract

A robust cell-free platform technology, ribosome display in combination with cloning, expression, and purification was utilized to develop single chain Fragment variable (scFv) antibody variants as pain therapy directed at the mouse cholecystokinin B (CCK-B) receptor. Three effective CCK-B peptide-specific scFvs were generated through ribosomal display technology. Soluble expression and ELISA analysis showed that one antibody, scFv77-2 had the highest binding and could be purified from bacterial cells in large quantities. Octet measurements further revealed that the CCK-B scFv77-2 antibody had binding kinetics of *K*_D_ = 1.794 × 10^–8^ M. Molecular modeling and docking analyses suggested that the scFv77-2 antibody shaped a proper cavity to embed the whole CCK-B peptide molecule and that a steady-state complex was formed relying on intermolecular forces, including hydrogen bonding, electrostatic force, and hydrophobic interactions. Thus, the scFv antibody can be applied for mechanistic intermolecular interactions and functional in vivo studies of CCK-BR. The high affinity scFv77-2 antibody showed good efficacy with binding to CCK-BR tested in a chronic pain model. In vivo studies validated the efficacy of the CCK-B receptor (CCK-BR) scFv77-2 antibody as a potential therapy for chronic trigeminal nerve injury-induced pain. Mice were given a single dose of the CCK-B receptor (CCK-BR) scFv antibody 3 weeks after induction of a chronic trigeminal neuropathic pain model, during the transition from acute to chronic pain. The long-term effectiveness for the reduction of mechanical hypersensitivity was evident, persisting for months. The anxiety- and depression-related behaviors typically accompanying persisting hypersensitivity subsequently never developed in the mice given CCK-BR scFv. The effectiveness of the antibody is the basis for further development of the lead CCK-BR scFv as a promising non-opioid therapeutic for chronic pain and the long-term reduction of chronic pain- and anxiety-related behaviors.

## 1. Introduction

Single-chain variable antibody fragment (scFv) antibodies are opening a new era of therapeutics, pharmacology, and pathophysiology research [1]. These technologies, used for over a decade as a cancer therapy, are overcoming previous challenges of providing therapeutic applications requiring central nervous system penetrance. Several scFvs antibodies are being investigated as therapeutics for arthritis, Creutzfeldt–Jakob, and Huntington’s disease due to their solubility, small size, and ability to cross the blood–brain barrier, unlike monoclonal antibodies (MAbs) available for migraine (Galcanezumab, Erenumab) [2,3,4]. These small, brain-penetrant antibodies are praised as having promising biotherapeutic applications for both the nervous and immune systems, now recognized as interactive in chronic pain. Despite the popularity of scFvs generated by ribosome display for immunotherapy, obtaining high-affinity scFvs from ribosome display libraries has remained a challenging task [5]. Here, we have developed a rapid generation of scFv antibodies against a small peptide fragment of the mouse receptor for neuropeptide cholecystokinin CCK-B (CCK2) by ribosome display (Figure 1). 

Understanding how an antibody interacts with its targets is critical for the development of that antibody as a therapeutic drug. Molecular docking [6,7] and molecular dynamic (MD) simulation [8] methods provide an advantageous means for studying the interaction between antigens and antibodies. Therefore, the present work has also studied predictions for the interaction between scFv77-2 antibody and CCK-B peptide by molecular docking and MD simulation. 

Validation of the efficacy of the lead CCK-BR scFv antibodies for the reduction of hypersensitivity, anxiety-, and depression-like behaviors was investigated in a chronic trigeminal neuropathic pain model in mice persisting 3–4 months. The CCK-BR scFvs directed to a mouse CCKBR fifteen amino acid peptide sequence were engineered with the ultimate goal of humanizing the antibodies for use in reducing dose requirement and tolerance of opioid analgesics for the treatment of chronic pain in patients.

CCK-BR is involved in several different aspects of the human pain experience that are particularly prominent in females [9]. CCKBR and its neuropeptide ligand, CCK, are widely expressed in the sensory ganglia, spinal cord, and brain pain circuitry [10,11]. Axotomy results in CCK upregulation in sensory neurons (30%) after 14 days [10]. A 4.7-fold upregulation of CCK-BR mRNA (*p* < 0.0001) is reported in a mouse sciatic nerve injury model [12]. Our microarray gene chip expression profile data identified >4-fold upregulation (*p* < 0.0001) of CCK-BR mRNA post day 3 in trigeminal ganglia (TG) compared to naïves in our chronic trigeminal neuropathic pain model [13]. CCK-BR mRNA remained upregulated 2.72-fold (*p* < 0.001) on post-day 21. In fact, CCKBR contributes to chronic pain in a variety of animal models, with gene expression changes over time [10,14,15]. 

Upregulation of CCK in primary sensory neurons is associated with morphine insensitivity in experimental neuropathic pain after sciatic nerve axotomy in the rat [10]. Abundant literature support is also available, casting CCK-B as a major player in anxiety, and panic disorder [16,17,18]. Block of the CCK-B receptor provides enhancement of morphine analgesia and opioid receptor tolerance [14,19,20,21,22,23,24,25,26,27,28,29,30]. More importantly, selective CCKBR antagonists enhance morphine analgesia and prevent/reverse tolerance without worsening respiratory depression in non-human primates and without side effects other than orthostatic dizziness in placebo-controlled clinical trials. Thus, CCKBR is an ideal candidate to impact both nociceptive and limbic components of chronic pain, identifying it as an important therapeutic target that as yet has no beneficial therapy available. 

## 2. Results

### 2.1. Anti-CCK-B scFv Antibodies Generated from Cell-Free Ribosomal Display

To determine whether the ribosomal display is suitable for the generation of antibodies against the mouse CCK-B receptor, we immunized mice with a custom 15-amino-acid extracellular mouse sequence CCK-B receptor peptide (Appendix A) [31]. CCK-B contains an extra loop of amino acids in the extracellular domain; which may serve as a target for immunotherapy. For ribosome displayed scFv antibody libraries, the immunoglobulin VH and VL regions joined to a 20-amino-acid flexible linker [(G4S)4] were constructed using cDNA, synthesized from RNA extracted from spleens of five mice (Figure 1, Appendix A), as described previously [32,33]. The amplified PCR product was the expected size of about 750 bp [31]. The final DNA template encoding the library flanked by a T7 site was used in an in vitro ribosome display with a single selection step with mouse CCK-B receptor peptide (Figure 1) [32]. 

The ribosome-displayed scFv library was panned against CCK-BR peptide with 3 rounds of selection, PCR products cloned into pGEM-T vector DNA, VH-VL transformants (up to 50 unique, diverse scFv clones) randomly selected, sequenced, and aligned using Clustal Omega. Their amino acid sequences were deduced, and three complementary determining regions (CDRs) and four framework (FW) regions were identified in each of the heavy (VH) and light (VL) chain fragments. A 20-amino-acid [(G4S)4] linker was also present. Following alignment with each other, significant diversity in the VH and VL chains was observed, especially in the CDRs. Variability was also noted in the framework regions. No two clones had identical VH or VL fragments. The aligned amino acid sequences of 7 clones using Clustal Omega from the library are shown in Figure 2. The framework regions (FRs) and CDRs were determined by the IMGT information system (https://www.imgt.org/IMGT_vquest/vquest?livret=0&Option=humanIg, accessed on 18 April 2023) [34]. The length of CDR1 VH, with an average length of 9 amino acid residues, CDR2 VH with an average length of 7 residues, CDR3 VH ranged from 9 to 13 amino acid residues, with an average length of 11 residues, CDR1 VL ranged from 5 to 12 with an average length of 11 residues, while 3 amino acid residues were found in CDR2 VL and 9 amino acid residues were found in CDR3 VL (Appendix A). Comparison of the heavy chain and light chain gene families of the isolated clones with VBASE2 Ig database (http://www.dnaplot.com, accessed on 18 April 2023) showed that VH and VL of these clones belonged to the mouse Ig heavy family VH2 and VH15 and light family IGKV2, IGKV8 and IGKV4/5, respectively (Appendix A).

The resulting isolation panel of 7 anti-CCK-B receptor recombinant antibodies was subcloned into a pET32a expression vector, expressed and purified from *E. coli* cytoplasm, and fractions were analyzed by SDS-PAGE (to confirm the integrity and purity) followed by Western blot and SEC-UPLC (to determine the aggregates), as carried out previously to generate scFvs against Zika virus and filovirus glycoproteins [32,33]. 

#### 2.1.1. Characteristics of the Lead CCKBR scFv 77-2 

The scFv77-2 exhibited a single band with an apparent molecular mass 26 kDa in gel electrophoresis and Western blot under denaturing and reducing conditions (Figure 3), which is within the expected size for the monomeric form of this protein. This antibody also maintains the required monomer resolution on SEC-UPLC (Figure 3). Endotoxin level in this protein was <1.0 EU/mg as determined by the LAL method. 

The scFvs were screened by ELISA for their specificity, affinity, and cross-reactivity. The seven scFvs demonstrated differential CCK-B receptor binding capability and specificity (did not cross-react with CCK-A and P2X4 receptor proteins) by indirect ELISA as shown in our previous study [31]. This assay was specific for CCK-B receptor peptide, as the negative control anti-EBOV scFv4-2 antibody did not react [33]. ScFv77-2, scFv14-3, and scFv134-1 had the highest, second, and third highest affinity, respectively, whereas others had a lower affinity, reflecting that the panning was efficient in selecting clones of high affinity [31]. All these antibodies bound to CCK-BR peptide in a concentration-dependent manner and showed high apparent affinity. Octet measurements further revealed that the CCK-B scFv77-2 antibody had binding kinetics of *K*_D_ = 1.794 × 10^–8^ M (Figure 4). Among three high-affinity CCK-B receptor scFv antibodies (14-3, 77-2 and 134-1), scFv77-2 was selected for testing behavioral functionality in vivo 3 weeks post nerve injury. This lead scFv is ~1/6 of an IgG and thus can access the central nervous system. 

#### 2.1.2. Protein Surface Analysis

Patch properties were computed from molecular surfaces projected at the water-probe distance (1.4 Å) away from the vdW surface of the protein. The protein surface patch calculation determines three classes of surface patches based on the respective hydrophobic and hydrophilic surface potential values: hydrophobic (green), positive (blue), and negative (red). Input structures were refined prior to protein surface patch calculation. The system pH was set at the appropriate value and atom charges were assigned according to the OPLS3.0 force field. AggScore was calculated on the set of three antibody structures with known liabilities. The score was able to predict their aggregation propensities in perfect rank order (Figure 5). A recent publication has reported the higher aggregation potential of antibodies discovered via phage display [35,36,37] and the associated negative correlation with clinical success. It is therefore important to prioritize antibodies based not only on affinity but also on those with low aggregation potential. Using an aggregation propensity algorithm, we calculated an aggregation score, AggScore, for scFv14-3 and scFv77-2 had an AggScore of 87.5 and 70.4 respectively, whereas scFv134-1 had a score of 50. Figure 5 illustrates the localization of these segments (mainly referring to hydrophobic stretches) in the scFv amino acid sequence. Most of the aggregation hotspots were predicted near the CDRs of V_H_ and V_L_ domains. 

#### 2.1.3. Prediction of scFv77-2 Interaction with CCK-BR Peptide

In silico molecular analyses provided insights into the interaction between the scFv77-2 and its ligand, CCK-B peptide, using for that a homology model of the scFv assembly in a monomeric closed state. Docking studies were performed with the PIPER protein-protein docking program in the BioLuminate product. 

The analysis of the scFv77-2 monomer-CCK-B peptide putative interface and the 3D cartoon representation of the docked complex is depicted in Figure 6. The molecular dynamics simulation confirmed the mouse scFv77-2-mouse CCK-B peptide docking prediction with sufficiently large and highly stable interface with amino acid side chain residues of scFv forming electrostatic/aromatic/hydrophobic interactions with CCK-B counterparts (Figure 6, Appendix A). The scFv77-2 module was predicted to contact the CCK-B peptide residues Glu40, Arg45, Arg50, and Glu53 (Figure 6). The scFv77-2 residues implicated in such binding are positioned mainly at VH-CDR1, VH-CDR3, and VL-CDR1, as indicated in Figure 6. These results are comparable with the two-dimensional and three-dimensional structure of scFv in complex with CCK-B peptide. The hydrogen bond interactions of the complex were elucidated to validate the binding of the scFv to the CCK-B peptide predicted by the docking simulation studies. The number of hydrogen bonds between the scFv and CCK-B peptide complex (acceptor/donor) was calculated and matched for identity with the hydrogen bond residues predicted in the docking analysis (Figure 6). The residues involved in hydrogen bonding during the post-simulation analysis of trajectories were found as the same residues contributing to hydrogen bonding during the docking analysis. Post-simulation MMGBSA analysis showed binding energy of −33.74 Kcal/mol.

### 2.2. In Vivo Validation

The CCK-BR scFv was efficacy tested in the mouse FRICT-ION (foramen rotundum inflammatory compression of the trigeminal infraorbital nerve) model of craniofacial chronic neuropathic pain. Chronic neuropathic pain was induced in BALB/c male and female mice (8 weeks old) using the easily induced but durable FRICT-ION model [38]. The model is described in Section 4.4.

Experimental timeline (Figure 1) indicates baseline and weekly von Frey mechanical sensitivity behavioral testing, surgical model induction, treatment time point, and testing of anxiety- and depression-like behaviors in weeks 8–10. The treatment with scFv antibodies is given either as a single intraperitoneal injection or intranasal application in week 3 unless otherwise noted. The * indicates two potential experiment endpoints for tissue harvest. The data presented below provide evidence of efficacy for the reduction of pain-related measures.

#### 2.2.1. Selection of the Lead CCK-BR scFv

Three CCKBR scFv antibodies with the highest binding affinity (77-2, 134-1, 14-3) were given as a single dose 3 weeks post-surgical model induction to determine efficacy (intraperitoneal, i.p., 4.0 mg/kg) as shown in our previous study [31]. The CCK-BR scFv 77-2 was chosen as the lead scFv of choice moving forward due to its best binding affinity and optimal reduction of hypersensitivity pain-related behaviors. 

#### 2.2.2. CCK-BR scFv for Evoked Hypersensitivity in Male and Female Mice

The dose study published previously included 0.04 mg/kg, 0.4 mg/kg, 4.0 mg/kg, and 40 mg/kg treatment 3 weeks post model induction [31]. A single CCK-BR scFv given in week 3 provided a delayed but durable alleviation tested through 10 weeks. A Zika scFv control was used to ensure that the positive effects were specific to the CCK-BR scFv and not an effect of the scFv alone. For both mechanical and cold hypersensitivity the higher doses were effective (0.4–40 mg/kg). The optimal dose selected was 4 mg/kg for subsequent studies.

Initially given as an intraperitoneal injection, the optimal dose (4 mg/kg) was given once in week 3 to neuropathic pain model male and female mice (n = 10, more than one group, more than one scFv batch) (Figure 7A,B). No male versus female differences were evident.

Intranasal administration of CCK-BR scFv was also effective. Evoked reflexive mechanical responses were tested at baseline and weekly after model induction. Single-dose (4 mg/kg, n = 4/dose) administration given intranasally (i.n., 6 μL) was equally effective compared to i.p. administration for alleviation of mechanical hypersensitivity (Figure 8A). Trigeminal nerve endings are abundant in the nasal cavity.

Cold hypersensitivity was also reversed by the CCK-BR scFv as shown (Figure 8B). Cold hypersensitivity was not evident in mice with FRICT-ION chronic pain model treated with 4 mg/kg doses or greater.

#### 2.2.3. Pre-Treatment Is Not Efficacious

Effective treatments were all given post-model induction. When the 4 mg/kg dose of CCK-BR scFv was given as a pre-treatment three weeks before induction of the FRICT-ION model, the scFv had no effect (Figure 9A). If the pre-treatment was followed by post-treatment at 3 weeks, effectiveness was similar to a single treatment given in week 3 alone (Figure 9A). Thus, there was no additive effect. When five daily treatments with scFv were given in week 3 there was no additional alleviation. Weekly (Figure 9B, green arrows and line) and biweekly (Figure 9B, black arrows and line) treatments may be more efficacious. In another case, there was no effect when prior treatment with the scFv was given before the acute surgical incision pain model. (Figure 9C).

#### 2.2.4. CCKB Pharmacologic Comparator LY225910

Mice with FRICT-ION were treated daily for 8 days with the pharmacologic comparator, the selective CCKB inhibitor LY225910 (10 mg/kg) or vehicle. The FRICT-ION model had been induced 7 weeks prior, and thus mice were fully hypersensitive. Mechanical threshold was tested with von Frey filaments each morning prior to the treatment (Figure 10). Attenuation of the hypersensitivity began by the day 5 treatment and was significantly persistent on days 6–8. These results found the efficacy was similar to the attenuation conferred by the single dose of CCKB scFv following a similar one week recovery time course. This suggests the mechanism of action for the scFv is also interference with the CCKB in the nociceptive system. The study provides support for potential use of the scFv for durable relief of orofacial hypersensitivity. Other future studies are required to confirm this indication.

#### 2.2.5. Efficacy of CCK-BR scFv for Prevention of Pain-Related Anxiety 

The lead 77-2 scFv was selected based on the pK assessment and the mechanical hypersensitivity trials determining the optimal administered dose was 4 mg/kg. Two different anxiety tests were employed to assess the effects of CCK-BR scFvs. CCK-BR scFvs given in week 3 diminished the anxiety-like behaviors that develop after 4–6 weeks of persistent hypersensitivity in FRICT-ION (or in any chronic model). Anxiety- and depression-like behaviors were tested once in chronic weeks 6–10 to avoid the practice effects reported with re-testing. Initially, a pilot test with 3 scFvs (14-3, 77-2, 134-1) for anxiety with the zero maze found only the 14-3 scFv was similar to naïve mice for distance traveled (Figure 11A). Both scFvs 77-2 and 14-3 were effective in maintaining time in the open areas of the zero maze (Figure 11B). Dose-dependent data for the zero maze assessing reduction of anxiety-like behavior found all doses of CCK-BR scFv 77-2 maintained distance traveled equivalent to naïve mice (Figure 11D). For a time in the open area of the zero maze the 3 highest doses of 77-2 scFv were effective for reducing that anxiety measure (Figure 11C). 

The dose dependency data for the light/ dark anxiety test for male mice are shown in Figure 12, as well as data for females with the lead scFv (Figure 11, Figure 12 and Figure 13). Untreated mice with FRICT-ION, mice treated with low dose 0.04 mg/kg CCK-BR scFv, and mice treated with the control Zika scFv all developed both anxiety and depression. The dose study in male mice indicated the higher doses (0.4, 4, and 40 mg/kg) did develop changes in rearing behaviors (Figure 12C), but not in light occupancy (Figure 12D) time measures. 

In the female mice, the single 4 mg/kg dose tested prevented the development of all of the anxiety-like measures, with only the vehicle-treated FRICT-ION mice displaying anxiety with the light/dark test (Figure 12E,F).

#### 2.2.6. Efficacy of CCK-BR scFv for Prevention of Depression

Dose-dependent prevention of depression-like behavioral data are provided for mice in Figure 13. All doses of the CCK-BR scFv 77-2 (0.04, 0.4, 4, and 40 mg/kg) prevented the depression-like behaviors seen in FRICT-ION model vehicle-treated mice. This included the decrease in the number of times groomed and the total grooming time, standard depression measures.

The equivalent data with the optimal 4.0 mg/kg dose also prevented these same depression-like behaviors in female mice.

#### 2.2.7. Preliminary Demonstration of Brain Penetrance

Our lead scFv 77-2 with highest affinity for CCKBR (Kd 195 nM, 750 bp) is ~1/6 of a Mab, half the molecular weight, and thus was predicted to access the central nervous system (CNS). The presence of the His-tag marker remaining in the trigeminal ganglia and amygdala tissue homogenate seven weeks after the single i.p. injection suggests the CCK-BR scFv 77-2 either crosses the blood brain barrier or can be transported by the nerve endings of the trigeminal nerve in the medullary dorsal horn to pain pathway components such as the amygdala (Figure 14). 

The presence of the His-tag was shown in the medulla in a previous study [31]. 

#### 2.2.8. Murine CCKBR scFv 77-2 Reduces DRG Neuron Excitability

To determine the effect of murine CCKBR scFv 77-2 on DRG neuron excitability, we performed whole-cell patch-clamp electrophysiology recordings of cultured DRG neurons at 18–40 h post-plating from naïve mice (*n* = 3, Figure 15A). We established that a 1 h pre-treatment in vitro with 10 ug/mL murine CCKBR scFv 77-2 produced a statistically significant (*p* < 0.01, one-way ANOVA with post hoc Tukey’s multiple comparisons test) in firing frequency compared to its vehicle control in response to stepwise current injection as shown in Figure 15B. In contrast, a 1 h pre-treatment with LY225910 (100 nM) did not produce a statistically significant reduction compared to its vehicle control, although firing frequency was reduced. There were no statistically significant differences observed in intrinsic electrophysiological properties (input resistance, resting membrane potential, or rheobase). 

## 3. Discussion

### 3.1. Generation of CCKBR scFv

Seven scFvs targeting CCK-BR generated using the robust platform technology were described here. This was accomplished with cell-free ribosome display in combination with cloning, expression, and purification of an anti-CCK-B scFv. Selection of a lead scFv antibody from the three with highest binding affinity allowed in vivo efficacy validation studies. Reduction of ongoing persisting pain was demonstrated with the lead CCK-BR scFv in in vivo studies with an orofacial neuropathic pain model.

Aggregation is a common problem affecting biopharmaceutical development that can have a significant effect on the quality of the product, as well as the safety of patients, particularly because of the increased risk of immune reactions [39]. The aggregation of the CCK-B scFv antibodies potentially reflects aspects related to the employed protein expression and refolding strategy. Large-scale production of scFvs in bacterial expression systems, although practical and time-efficient, often leads to a product containing aggregates [40]. This may be significantly boosted by the particular propensity of the designed scFv to aggregate. Our in silico analyses showed some aggregation hotspots within the lead CCK-B scFvs amino acid sequences, but also those lead scFvs with low aggregation potential. Furthermore, our computational approaches are suitable during early drug development to select lead scFv molecules with reduced risk of aggregation and optimal developability properties. 

Our exploratory in-silico analyses additionally provide mechanistic insights into the antigen–antibody interaction. scFv77-2 was predicted to contact non-linear stretches within the CCK-B surface. In silico analysis of the putative scFv-CCK-B interface revealed that most of the antibody determinants involved in antigen recognition are located within the heavy-chain CDR1, heavy-chain CDR3, and the light-chain CDR1, whose residues are less prone to aggregation.

### 3.2. In Vivo Efficacy of CCKBR scFv

Thus, current treatment with analgesics even when combined with antidepressants and/or anticonvulsants are generally unsatisfactory in providing pain relief [41]. As an example of the unmet need, the current poor response rate to analgesics for painful trigeminal neuropathy among women for providing >50% reduction of pain intensity is only 11% [41]. The chronic pain experience exerts powerful persisting influences on the brain, inducing permanent circuitry alterations that diminish physical and mental function. Effective non-addictive, non-opioid therapeutics for chronic orofacial pain remain a critical need. 

Previously, a CCK octopeptide antagonist (CCK-8) was reported to suppress binding of naloxone to opioid receptors [42]. The study suggested the CCK-8 might be (1) suppressing opioid binding by uncoupling opioid receptors from their G-protein effectors pre-synaptically and (2) reducing the number and affinity of opioid receptors through a preventative post-receptor mechanism. Further development of the CCK-BR scFv will provide its relevance for use not only in regard to chronic neuropathic pain but its potential adjuvant use to reduce opiate dose and tolerance. CCK-BR is involved in several different aspects of the human pain experience that are particularly prominent in females [9]. CCKBR and its neuropeptide ligand, CCK, are widely expressed in sensory ganglia, spinal cord, and the brain pain circuitry [10,11]. Axotomy results in CCK upregulation in sensory neurons (30%) after 14 days [10]. A 4.7-fold upregulation of CCK-BR mRNA (*p* < 0.0001) is reported in a mouse sciatic nerve injury model [12]. Our microarray gene chip expression profile data identified >4-fold upregulation (*p* < 0.0001) of CCK-BR mRNA post day 3 in trigeminal ganglia (TG) compared to naïves in our chronic trigeminal neuropathic pain model [13]. CCK-BR mRNA remained upregulated 2.72-fold (*p* < 0.001) on post day 21. In fact, CCKBR is contributory to chronic pain in a variety of animal models, with gene expression changes over time [10,14,15]. 

Engineered antibodies of this type feature binding activity similar to monoclonal antibodies but with stronger affinity and thus are suitable for in vivo models. The increased tissue penetrability due to their smaller size provides access to the peripheral nerve ganglia and the pain pathway sites centrally demonstrated with Western blots. More importantly, scFv antibodies have promising biotherapeutic applications for both nervous and immune systems, now recognized as interactive in chronic pain. The scFv antibodies have higher affinity, stability, solubility, and binding specificity for cholecystokinin B but not A receptor. The scFv optimized with the best binding affinity were selected for the in vivo and in vitro efficacy demonstrated in the studies presented. It is well known that cholecystokinin B receptor and its neuropeptide ligand are upregulated in chronic neuropathic pain and stress models. They are abundant throughout pain pathway sites.

### 3.3. Efficacy of CCKBR scFv on Mechanical and Cold Hypersensitivity

The in vivo study for CCKBR scFv demonstrated efficacy for reduction of mechanical hypersensitivity. The tests found the CCKBR scFv was equally effective in both sexes with all doses (0.4–40 mg/kg), with no side effects, loss of weight, or change in organ weight. Cold hypersensitivity was increased by the FRICT-ION model and diminished by CCK-BR scFv 77-2 (0.4, 4 and 40 mg/kg). The PBS vehicle, Zika scFv, and low dose CCK-BR scFv 77-2 (0.04 mg/kg) were ineffective. Statistically significant reduction of both mechanical and cold hypersensitivity was durable, shown here persisting 7 weeks after a single i.p. or i.n. dose. 

### 3.4. Efficacy for Anxiety- and Depression-like Behavior

Remarkably, the hypersensitivity persists in untreated mice with the FRICT-ION chronic neuropathic pain model through 100 days. In addition to being a great model of craniofacial neuropathic pain, FRICT-ION is the only model of trigeminal neuralgia since it is responsive to carbamazepine [43]. The reduction of hypersensitivity at 1 week prevents the development of many of the anxiety and depression measures tested here with the zero maze and the light/dark tests. Anxiety behaviors arise in the untreated mice with FRICT-ION at 4–6 weeks and continue through the 10-week study. The accompanying depression, tested with the sucrose splash test, follows a similar time course in the untreated mice with FRICT-ION. Many of the depression-like behaviors are ameliorated by the single dose CCK-BR scFv treatment.

The diminished hypersensitivity prevents cognitive disruption seen in the mice with persisting hypersensitivity [31]. We have previously published results for the novel object cognitive measure mice with FRICT-ION that were significantly affected, while those receiving scFv 77-2 had results similar to naïve mice [31]. Likewise, results for the conditioned place preference box demonstrated that lead CCK-BR scFv 77-2 has no abuse potential [31]. 

The startling factor about the treatment with the small ribosome generated scFv antibodies is that they are effective with only a single dose in this chronic craniofacial pain model. The recovery is durable in this and the chronic spared nerve injury model [31]. The effectiveness of the small non-opiate scFv antibody targeting the cholecystokinin B receptor (CCK-BR) alleviation of chronic orofacial hypersensitivity is sufficient to prevent development of anxiety and depression in the chronic model.

### 3.5. Effect of scFv 77-2 on DRG Neuron Excitability

The direct effect of scFv 77-2 on reducing excitability of mouse trigeminal ganglia neurons has previously been demonstrated in Westlund et al., 2021 [31]. The data generated here agree with these findings and extend to DRG neurons as well. In addition, we show that LY225910, a commercially available CCK2 antagonist does not produce a statistically significant reduction in firing frequency in contrast to scFv 77-2. This is in agreement with in vivo findings that LY225910 needs to be administered daily for at least 1 week in order to produce an anti-allodynic effect, whereas only a single dose of scFv 77-2 is required. This also suggests that the acute effect on peripheral sensory neurons may be the differentiating factor that explains the mechanism of action of scFv 77-2’s relief of pain in chronic models compared to pharmacologics such as LY225910.

These findings support the use of single-chain Fragment variable antibodies generated with ribosome display technology as preferred non-opioid therapy to target and block the cholecystokinin B receptor in vivo and in vitro chronic neuropathic pain models. Future translational studies are needed to bring effective humanized scFv toward human use.

## 4. Materials and Methods

### 4.1. Generation of Cholecystokinin B (CCK-BR) scFvs

The methods of the immunization of mice, panning combinatorial antibody library against CCK-B peptide antigen using in vitro ribosome display, construction of antibody libraries, pull-down and selection, expression, purification, and characterization of antibodies have been described in the Appendix A.

### 4.2. Protein Surface Analysis 

Homology models of the scFv from amino acid sequences were generated by using the I-TASSER (**I**terative**T**hreading**ASSE**mbly**R**efinement) [44,45,46]. The predicted structural models were first refined using the protein preparation wizard [47] in Schrödinger’s BioLuminate suite prior to protein surface patch calculation. The Protein Surface Analyzer Tool, in combination with the aggregation score, AggScore, as defined by Sankar et al. [36] and implemented in the Schrödinger’s Biologics Suite was used to calculate the aggregation propensity of the selected scFv. The method used the three-dimensional structure to estimate the distribution of hydrophobic and electrostatic patches on the surface of the protein. 

### 4.3. Computational Modeling

The advanced computational protocol used for determining interactions between mouse scFv77-2 antibody and mouse CCK-B peptide involves several steps.

#### 4.3.1. Homology Modeling

The I-TASSER is a bioinformatics method for predicting 3D structure model of protein molecules from amino acid sequences [44,45,46,48]. The predicted structural models were validated using high-resolution protein structure refinement (Protein refinement module, Schrodinger-Prime module, Biologics suite, Schrodinger 2021-2) [49], ModRefiner [50], and fragment-guided molecular dynamics (FG-MD) simulation [51]. The protein and peptide were prepared using the Protein Preparation Wizard tool included in Maestro (Schrodinger Suite 2022-4). Water molecules, co-factors of crystallization, and ligands were removed, missing atoms were added, side chains and loops were filled by Prime, and hydrogens were added with Epik module options provided in the protein preparations wizard at physiological pH. This final structure of the protein was minimized with the OPSL-3e force field as implemented in Maestro with an implicit solvent (water). The final minimized structure was used for docking purposes. 

#### 4.3.2. Molecular Docking 

The refined models were docked according to the Fast Fourier Transform (FFT)-based program and PIPER [52]. The mouse scFv77-2-mouse CCK-B peptide docking were modeled using the PIPER protein–protein docking program in the BioLuminate product [53,54]. The largest cluster size with minimal local energy and a near-native state of the protein conformation was chosen. Docking results were validated using the LigPlot tool of Schrödinger suite 2022-4 or Ligplot^+^ v.2.2 software. An interactive map was studied to identify the chemical nature of the interactions such as hydrogen bonds, π–π interaction, side-chain bond, and backbone hydrogen bonds. Ligand–protein interaction maps were also used to predict the position and interacting amino acids of the scFv77-2 and CCK-B peptide. 

#### 4.3.3. Molecular Dynamics (MD) Simulations 

MD simulation studies for the selected docked poses were carried out by the Desmond module of Schrödinger suite 2022-4 with OPLS4 force field [55]. The protein–ligand complex was embedded in a predefined TIP3P water model in the orthorhombic box [56]. The box volume was minimized, and the overall system charge was neutralized by adding Na^+^ or Cl^−^ ions and 0.15 mM NaCl to construct near-physiological conditions. The temperature and pressure were kept constant at 300 K and 1.01325 bar throughout the simulation using Nose-Hoover thermostat [57] and Martyna-Tobias-Klein barostat [58] methods. The simulations were performed for >100 ns using NPgammaT ensembles for proteins and membranes ensemble considering the number of atoms, pressure, and timescale [59]. During simulations, long–range electrostatic interactions were calculated using Particle–Mesh–Ewald method [60] and the whole ensemble was constructed as a rigid body packing and relaxed gradually at 1.2 kilojoule of energy during the simulations [59]. The amino acid energy contributions that will be obtained from the prime molecular mechanics-generalized born surface area (MM-GBSA) calculation was used in our study to elucidate the key amino acids predicted to be critical protein–protein interaction. 

### 4.4. Surgical Induction of the Trigeminal Neuropathic Pain Model

Validation of in vivo efficacy was assessed in BALB/c male and female mice injected with the CCKBR scFv 3 weeks after induction of a chronic neuropathic pain model. The chronic model causing compression and chemical irritation of the trigeminal nerve is referred to by the acronym FRICT-ION (foramen rotundum inflammatory compression of the trigeminal infraorbital nerve) [38]. The FRICT-ION trigeminal neuropathic pain model was induced by inserting 3 mm of chromic gut suture (4-0) through a tiny scalpel incision in the oral buccal/cheek crease along the trigeminal maxillary nerve branch (V2) as it passes into the foramen rotundum of the skull. The surgery was performed in <10 min, including anesthetic induction and recovery time. In sham surgery group mice, the oral buccal incision was made but the nerve is untouched. Naïve control mice remained untouched but were subjected to all behavioral testing. Mechanical and cold hypersensitivity in FRICT-ION mice developed reliably on the snout in all animals within the next week. The experimental timeline is provided in the “Section 2.2. In vivo validation”.

### 4.5. Lead CCK-BR scFv Determination

Three CCK-BR scFv antibodies (77-2, 134-1, 14-3) with the highest binding affinity were given a single dose to determine efficacy. The lead scFv 77-2 with the highest affinity for CCKBR (*K*_D_ 195 nM, 750 bp) is ~1/6 the size of a MAb and thus can access the central nervous system (CNS). 

### 4.6. In Vivo Behavioral Read-Outs

The effectiveness of the CCK-BR scFv was assessed with methods standard in the field. Mean experimental results were compared among groups. Males and females were compared separately and together since there was no difference. This includes evoked and spontaneous behaviors that are relevant to the human condition and thus are better predictors of anti-allodynic, anxiety- and depression-like behavioral efficacy.

#### 4.6.1. Von Frey Fiber Assessment of Hypersensitivity

Assessment of sensitivity on the snout was performed before nerve injury to determine baseline threshold and performed weekly after nerve injury, through 10 weeks. Hypersensitivity is assessed by reflexive withdrawal response time to mechanical stimulation on the snout with graded thin nylon von Frey filaments with defined bending forces (tensile strength) (Figure 3D,E, Figure 4 and Figure 5) [38,61,62,63]. A trial consisted of 5 applications of several selected mid-range von Frey filaments applied once every 3 to 4 s. If no positive response was evoked, the next stronger filament was applied. The mean occurrence of withdrawal events in each of the trials was expressed as the number of responses out of 5: 0 indicates no withdrawal and 5 indicates the maximum number of withdrawals. An arithmetic algorithm was used to convert the fiber strength into grams force 30 when three of five responses were evoked from a given fiber. Behavioral assessment of hypersensitivity continued through 10 weeks post model induction. Responses to decreased gram force compared to controls indicated decreased sensitivity threshold or “hypersensitivity”. 

#### 4.6.2. Cognition Dependent Anxiety-like and Depression-like Behavioral Tests

Effectiveness of the CCK-BR scFv to anxiety- and depression-related behaviors are assessed using the light/dark box, elevated zero maze, and sucrose splash tests. Anxiety- and depression-like behaviors are quantified prior to euthanasia in week 10 [38,64,65]. Computer-linked video recordings are used to quantify the behaviors. 

#### 4.6.3. Light/Dark Place Preference Test

The light/dark box used to assess anxiety-related behaviors consists of two equally sized chambers, one darkened and one brightly illuminated. Collected variables in this two-chamber test are (1) the time spent in each chamber, (2) the number of transitions between chambers, (3) the number of rearing events, (4) entry latency into the light chamber, and (5) latency of first re-entry (transition) back into the dark chamber. Anxiety-like behavior is significantly greater in neuropathic pain models that do not receive the parent CCKBR scFv (*n* = 6, * *p* < 0.05 ANOVA). 

#### 4.6.4. Elevated Zero Maze

The elevated plus maze is a widely used test for measuring anxiety-like behavior, by determining a preference between a comparatively safe environment (closed arms) and a threatening environment (open arms). In principle, the more “anxious” the subjects are, the less likely they will explore a risky or threatening environment. Anxiety-like behavior in the elevated zero maze is determined by the (1) number of open and closed quadrant entries, (2) total open and closed area occupancy, and (3) by the number of exploratory rearing events. High anxiety states are directly related to open area avoidance. 

#### 4.6.5. Sucrose Splash Test

Depression-like behavior is validated with the sucrose splash test where measurement of decreased grooming behavior is a symptom of depression. Frequency, duration, and latency of grooming are scored (10 min) after spraying a 10% sucrose solution (~250 μL) on the base of the tail. Grooming time after sucrose splash test was increased significantly after scFvs 12 and 95 in an initial test (*n* = 6, * *p* < 0.05 ANOVA). 

### 4.7. Western Blot

Our lead scFv 77-2 with the highest affinity for CCKBR (*K*_D_ 195 nM, 750 bp) is ~1/6 the size of an IgG and thus can access the central nervous system (CNS), as shown with evidence of the His-tag marker remaining in the TG and amygdala 7 weeks after a single intraperitoneal injection of CCK-BR scFv 77-2.

### 4.8. Dorsal Root Ganglion Cultures

Animals were deeply anesthetized with 3% isoflurane and then decapitated prior to dissection of dorsal root ganglia (DRG) for primary cultures. DRG cultures were prepared as described previously in Malin et al. [66].

### 4.9. Whole-Cell Patch Clamp Electrophysiology 

Neurons were identified by infrared differential interference contrast (IR-DIC) connected to an IR2000 camera (Dage MTI, Michigan City, IN, USA). Current-clamp recordings were performed using a Molecular Devices Multiclamp 700B (Scientifica, Uckfield, UK). Signals were filtered at 5 kHz, acquired at 50 kHz using a Molecular Devices 1550B converter (Scientifica, UK), and recorded using Clampex 11 software (Molecular Devices, Scientifica, UK). Electrodes were pulled with a Zeitz puller (Werner Zeitz, Martinsreid, Munich, Germany) from borosilicate thick glass (GC150F, Sutter Instruments, Novato, CA, USA). Electrode resistance was 5–8 MΩ. Bridge balance was applied to all recordings. For DRG culture recordings intracellular solution contained (in mM) 125 K-gluconate, 6 KCl, 2 CaCl_2_, 10 HEPES, 10 EGTA, 2 Mg-ATP, pH 7.3 with KOH. Artificial cerebrospinal fluid (aCSF) contained (in mM) 113 NaCl, 3 KCl, 25 NaHCO_3_, 1 NaH_2_PO_4_, 2 CaCl_2_, 2 MgCl_2_, and 11 D-glucose. For brain slice recordings intracellular solution contained (in mM) 120 K-gluconate, 11 KCl, 1 CaCl_2_, 2 MgCl_2_ 10 HEPES, 11 EGTA, 4 Mg-ATP, 0.5 Na-GTP pH 7.3 with KOH. aCSF contained (in mM) 113 NaCl, 3 KCl, 25 NaHCO_3_, 1 NaH_2_PO_4_, 2 CaCl_2_, 2 MgSO_4_, HEPES 5 mM and 11 D-glucose. 

### 4.10. Statistical and Data Analysis

Behavioral data analysis was performed using GraphPad Prism (9.2.0). A *p*-value < 0.05 was considered statistically significant. Statistical tests are shown in the figure legends.

Electrophysiology data analysis was performed using Easy Electrophysiology (v.2.5.1) and Clampfit 11.2. Recordings were not corrected for junction potential. Experimenters were blinded during experiments and analysis. All statistical analysis was performed using GraphPad Prism (v9.2.0). A *p*-value < 0.05 was considered statistically significant. Statistical tests are shown in the figure legends.

## 5. Conclusions

While acute and post-surgical pain are effectively managed by opiates, the generation of therapies effective for persisting and chronic pain have been stymied by lack of understanding of the differences between the physiological and molecular characteristics of acute versus chronic pain now emerging. Chronic pain can induce permanent brain circuitry alterations that further diminish physical, emotional, and mental function. Urgently needed are non-opioid therapeutics that address and/or prevent the effects of chronic pain on higher brain processes such as anxiety and depression, without affecting cognitive functions. This unmet need is remedied with our CCK-BR scFv generated using ribosomal display technology, cloning, expression, and affinity purification described here. Three top scFv leads of eight generated were characterized in vivo and shown to be effective in reducing mechanical and cold hypersensitivity. More importantly, the CCK-BR scFv was able to stem the increase in anxiety and depression characteristic of the chronic trigeminal neuropathic pain model. Speculation as to whether this scFv targeting a mouse peptide sequence of mouse CCK-BR peptide is as effective or more effective in a human test system remains to be determined.

## 6. Patents

Therapeutic Antibody Fragments, Methods Of Making, And Methods of Use. US Patent WO-2020092883-A1, Publication date: 5 July 2020.

Therapeutic Antibody Fragments, Methods of Making, and Methods of Use. US Patent Application Pub. No. US 2021/0340265 A1, Authorized by Karin Westlund High, Ravi Venkata Durvasula, Adinarayana Kunamneni. Application No. 17/284,208, filed 9 April 2021, published 4 November 2021.

## Data Availability

Not applicable.

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
