# Peer review of "Rapid Generation and Molecular Docking Analysis of Single-Chain Fragment Variable (scFv) Antibody Selected by Ribosome Display Targeting Cholecystokinin B Receptor (CCK-BR) for Reduction of Chronic Neuropathic Pain"

_ijms, 2023, doi:10.3390/ijms241311035_

Round 1
Reviewer 1 Report
Overall, the manuscript is well written with a clear experimental design.
Just a few minor comments below:
89 - pixelated figure.
Most figures in this manuscript are blurry/pixelated. It would be beneficial to replace them with higher resolution versions for visual clarity
134-136 - misaligned title and pixelated table.
192, 286, 289, 291, 591 – name mismatch in figure reference (supplementary fig 123 vs scheme 123)
215 - figure title missing
396 - they are under Appendix A in 580-760. Move Appendix A into supplementary material section?
Author Response
Response to Review:
89 - pixelated figure.
Response: We have replaced the figure with new one.
Most figures in this manuscript are blurry/pixelated. It would be beneficial to replace them with higher resolution versions for visual clarity
134-136 - misaligned title and pixelated table.
Response: We have placed the title and table properly.
192, 286, 289, 291, 591 – name mismatch in figure reference (supplementary fig 123 vs scheme 123)
Response: We have corrected this in the text.
215 - figure title missing
Response: We have included figure legend.
396 - they are under Appendix A in 580-760. Move Appendix A into supplementary material section?
Response: We have moved Appendix A into supplementary material section.
Reviewer 2 Report
Line 17: wording needs to be improved: the affinity of the scFv was not engineered, instead suitable scFv were discovered
Line 19: what is meant by the word “reactivity”?
Line 25: Data of only one scFv have been shared, so the statement “the affinity increased the efficacy in vivo” is not valid
Line 33: What is meant by the statement “Dose-response data are provided for each validation”?
Line 52: In what respect are scFv used for chemotherapy?
Line 53: The statement “we have developed rapid generation of scFv antibodies by ribosome display” needs more clarity.
Line 89: test in figure is not readable.
Line 94: vaccinate implies immunization for therapeutic purpose which is not the case here.
Line 105: Assessing 50 clones is unusual low. Can you comment?
Line 126: please provide data on Western blot and SEC data for scFv.
Line 141: 85% monomer content is low; in particular, the % monomer is too low for any in vivo studies. Have the in vivo studies conducted with this % monomer?
Line 147: please ELISA data.
Line 152: please provide kon, koff of scFv77-2.
Line 155: please provide references and discussion on accessibility of CNS by scFv (in the actual discussion). Please provide cell binding data.
Line 162ff: What is the rationale, benefit and conclusion of the protein surface analysis, in particular of the AggScore, given that you already know that the scFv77-2 is aggregating?
Line 182ff: Please provide an estimate of the quality and accuracy of the interaction prediction. Do you have actual experimental data to ack-up the predictions? How does the prediction impact the pharmacology?
Line 215: figure legend is missing.
Line 223: please provide data.
Line 233ff: reference to figure 7B is missing.
Line 293: scFv is 1/6 of an IgG.
Line 299: The transfer of the scFv into the CNS is pivotal and should be backed-up by orthogonal studies such as IHC including isotypes controls.
Line 341: scFv have not been refolded experimentally in this study.
Line 355-390: this chapter is a long literature review and should part of the introduction and can be shorted. The discussion lacks the discussion of the in vivo studies completely.
Line 392ff: Data and methods are not included in the supplementary material as claimed in this paragraph.
Author Response
Response to Review:
Line 17: wording needs to be improved: the affinity of the scFv was not engineered, instead suitable scFv were discovered
Response: Yes, I agree with the reviewer that the affinity of the scFv was not engineered, instead suitable scFv was discovered. We have corrected this sentence in the manuscript.
Line 19: what is meant by the word “reactivity”?
Response: We mean the word “reactivity” is “binding” and corrected this word in the manuscript.
Line 25: Data of only one scFv have been shared, so the statement “the affinity increased the efficacy in vivo” is not valid
Response: We have corrected this statement in the manuscript.
Line 33: What is meant by the statement “Dose-response data are provided for each validation”?
Response: we have removed this sentence from the manuscript.
Line 52: In what respect are scFv used for chemotherapy?
Response: scFv is for immunotherapy, not for chemotherapy. We have corrected this in the manuscript.
Line 53: The statement “we have developed rapid generation of scFv antibodies by ribosome display” needs more clarity.
Response: We have rewritten this sentence in the manuscript.
Line 89: test in figure is not readable.
Response: We have included text in the figure, now it is readable.
Line 94: vaccinate implies immunization for therapeutic purpose which is not the case here.
Response: Yes, we have immunized mice to generate antibodies, not for therapeutic purpose.
Line 105: Assessing 50 clones is unusual low. Can you comment?
Response: We usually assess 10 to 50 unique, diverse scFvs
Line 126: please provide data on Western blot and SEC data for scFv.
Response: We have provided this data in the manuscript.
Line 141: 85% monomer content is low; in particular, the % monomer is too low for any in vivo studies. Have the in vivo studies conducted with this % monomer?
Response: We have SE-UPLC data for purified antibody from another batch (used for in vivo studies) and included this data in the manuscript.
Line 147: please ELISA data.
Response: We have shown ELISA in our previous study (Westulund et al. 2021)
Line 152: please provide kon, koff of scFv77-2.
Response: We have included kon, koff of scFv77-2 data in the Fig.
Line 155: please provide references and discussion on accessibility of CNS by scFv (in the actual discussion). Please provide cell binding data.
Response: We have discussed on the accessibility of CNS by scFv in the section “2.2.7.”
Line 162ff: What is the rationale, benefit and conclusion of the protein surface analysis, in particular of the AggScore, given that you already know that the scFv77-2 is aggregating?
Response: The results demonstrate that the generated scFv77-2, although prone to aggregation, comprises an active anti-CCK-BR product that contains monomers and small oligomers.
Line 182ff: Please provide an estimate of the quality and accuracy of the interaction prediction. Do you have actual experimental data to ack-up the predictions? How does the prediction impact the pharmacology?
Response: Functionally, the scFv77-2 preparations specifically recognize CCK-BR and reduce chronic pain- and anxiety-related behaviors in an experimental model. In silico molecular analysis provided insights into the aggregation propensity and the antigen recognition by scFv units. Antigen-binding determinants were predicted outside the most aggregation-prone hotspots. Overall, our experimental and prediction dataset describes an scFv scaffold for the scFv77-2 and also provides insights to further engineer non-aggregated anti-CCK-B scFv-based tools for therapeutic and research purposes.
Line 215: figure legend is missing.
Response: We have provided figure legend in the manuscript.
Line 223: please provide data.
Response: We have provided the data in the manuscript
Line 233ff: reference to figure 7B is missing.
Response: We have included a reference for figure 7B.
Line 293: scFv is 1/6 of an IgG.
Response: We have included this in the manuscript.
Line 299: The transfer of the scFv into the CNS is pivotal and should be backed-up by orthogonal studies such as IHC including isotypes controls.
Response: We agree with the reviewer about this and it was in our previous study.
Line 341: scFv have not been refolded experimentally in this study.
Response: Our scFv was successfully obtained in a soluble form from E. coli (permit the formation of stable disulphide bonds within the cytoplasm) cytoplasm. Yes, scFv has not been refolded experimentally in this study.
Line 355-390: this chapter is a long literature review and should part of the introduction and can be shorted. The discussion lacks the discussion of the in vivo studies completely.
Response: We have shortened this text and included the discussion of the in vivo studies in the manuscript.
Line 392ff: Data and methods are not included in the supplementary material as claimed in this paragraph.
Response: We have included this in the supplementary material.